# Activation of a Sweet Taste Receptor by Oleanane-Type Glycosides from *Wisteria sinensis*

**DOI:** 10.3390/molecules27227866

**Published:** 2022-11-15

**Authors:** Samir Hobloss, Antoine Bruguière, David Pertuit, Tomofumi Miyamoto, Chiaki Tanaka, Christine Belloir, Marie-Aleth Lacaille-Dubois, Loïc Briand, Anne-Claire Mitaine-Offer

**Affiliations:** 1Centre des Sciences du Goût et de l’Alimentation, CNRS, INRAE, Institut Agro, Université de Bourgogne, Franche-Comté, 21000 Dijon, France; 2Graduate School of Pharmaceutical Sciences, Kyushu University, Fukuoka 812-8582, Japan

**Keywords:** *Wisteria sinensis*, Fabaceae, triterpene glycosides, sweet taste, TAS1R2/TAS1R3

## Abstract

The phytochemical study of *Wisteria sinensis* (Sims) DC. (Fabaceae), commonly known as the Chinese Wisteria, led to the isolation of seven oleanane-type glycosides from an aqueous-ethanolic extract of the roots. Among the seven isolated saponins, two have never been reported before: 3-*O*-α-L-rhamnopyranosyl-(1→2)-β-D-glucopyranosyl-(1→2)-β-D-glucuronopyranosyl-22-*O*-acetylolean-12-ene-3β,16β,22β,30-tetrol, and 3-*O*-β-D-xylopyranosyl-(1→2)-β-D-glucuronopyranosylwistariasapogenol A. Based on the close structures between the saponins from *W. sinensis*, and the glycyrrhizin from licorice, the stimulation of the sweet taste receptor TAS1R2/TAS1R3 by these glycosides was evaluated.

## 1. Introduction

Since the end of the 20th century, obesity has been one of the biggest health problems worldwide and has been linked in the long term to many different diseases such as type 2 diabetes, cardiovascular disease, hypertension, metabolic syndrome, and dyslipidemia [1,2]. However, reducing sugar intake can be very difficult, knowing that people’s taste preference for sweetness is innate [3]. Thus, to limit the prevalence of diseases linked to this excessive consumption of sugar, the search for natural non-caloric compounds with a sweet taste and the development of artificial sweeteners has increased [4]. Some natural triterpene glycosides activate sweet taste receptors, such as glycyrrhizin from licorice (*Glycyrrhiza glabra* L., Fabaceae), which stimulates the heterodimer TAS1R2/TAS1R3 [5,6]. The Fabaceae family is one of the most important families of Angiosperms, being the third largest of this group with 727 genera and about 19,325 species [7]. From a chemotaxonomic point of view, the two genera *Wisteria* Nutt. and *Glycyrrhiza* L. belong to the Faboideae subfamily, so they are phylogenetically very close [8]. Moreover, oleanane-type glycosides isolated from *Wisteria* species like *W. frutescens* [9], *W. floribunda* cultivars [10], and *W. brachybotrys* [11], showed some structural similarities with glycyrrhizin [12]: a β-carboxyl group at position C-30, a ketone function, and a 3-*O*-β-D-glucuronopyranosyl moiety.

Therefore, the aim of this paper is the isolation and the structural analysis of oleanane glycosides from the Chinese Wisteria,* W. sinensis* (Sims) DC., as well as the evaluation of their sweet taste by cell-based receptors assays. Seven saponins were purified after an extraction of the roots, and their structures were established. Two previously undescribed (**1**,**2**) and five known ones (**3**–**7**) (Figure 1) were reported, as 3-*O*-α-L-rhamnopyranosyl-(1→2)-β-D-glucopyranosyl-(1→2)-β-D-glucuronopyranosyl-22-*O*-acetylolean-12-ene-3β,16β,22β,30-tetrol (**1**), 3-*O*-β-D-xylopyranosyl-(1→2)-β-D-glucuronopyranosylwistariasapogenol A (**2**), wistariasaponin A (**3**) [13], wistariasaponin D (**4**) [11], dehydroazukisaponin V (**5**) [14], astragaloside VIII (**6**) [15], and azukisaponin V (**7**) [14]. Compounds **1**–**3** have been tested on the TAS1R2/TAS1R3 receptor to establish structure–activity relationships.

## 2. Results and Discussion

The saponins **1**–**7** were isolated from an aqueous-ethanolic extract of the roots of *W. sinensis* by various solid/liquid chromatographic methods, vacuum liquid chromatography (VLC), medium pressure liquid chromatography (MPLC), on normal and reverse phase RP-18 silica gel, and size exclusion chromatography on Sephadex LH-20.

The mass spectrum of compound **1** in HRESIMS (positive mode), reveals a quasi-molecular ion at *m*/*z* 1023.5051 [M + Na] ^+^, in agreement with the molecular formula C_50_H_80_O_20_Na. This suggests a molecular mass of 1000 g/mol.

The structure of the aglycone was determined using 1D and 2D NMR spectra, mainly COSY, TOCSY, ROESY, HSQC and HMBC (Table 1).

The ^1^H NMR spectrum part of the aglycon displayed signals attributable to seven angular methyl groups (3H, s, each) at δ_H_ 0.79 (H-28), 0.88 (H-23), 0.91 (H-29), 0.98 (H-25), 1.02 (H-26), 1.11 (H-24) and 1.26 (H-27), one olefinic proton at δ_H_ 5.33 (1H, br t, *J* = 3.0 Hz, H-12), three secondary alcoholic functions at δ_H_ 3.22 (1H, H-3), 4.15 (1H, dd, *J* = 11.7, 4.7 Hz, H-16) and 5.24 (1H, dd, *J* = 3.5, 1.2 Hz, H-22) and a primary one at δ_H_ 3.49 (1H, d, *J* = 14.1 Hz, H-30_a_), 3.51 (1H, d, *J* = 14.1 Hz, H-30_b_). Additionally, an HMBC correlation at δ_H_ 2.02 (3H, s)/δ_C_ 171.1 revealed the presence of an acetyl group. A classical 3-*O*-glycosidic linkage can be observed according to the deshielded signals of position 3 at δ_H_ 3.22 (1H, H-3)/δ_C_ 90.4 (C-3). Moreover, a HMBC cross-peak at δ_H_ 4.15 (1H, H-16 _ax_)/δ_C_ 13.8 (C-28), and a COSY correlation between 4.15 (1H, H-16_ax_) and δ_H_ 1.31 (1H, H-15_eq_), 1.76 (1H, H-15_ax_), allowed the localization of a secondary alcoholic function at the C-16 position. A deshielded signal of a proton at δ_H_ 5.24 suggests an acylation by an acetyl group at δ_H_ 2.02 (3H, s)/δ_C_ 171.1. The localization of this *O*-acetyl group at the 22-position was determined by an HMBC cross-peak between δ_H_ 0.79 (3H, H-28) and δ_C_ 74.2 (C-22), and a COSY correlation between δ_H_ 5.24 (1H, H-22_eq_) and 1.49 (1H, H-21_a_), 1.75 (1H, H-21_b_). The CH_2_OH group was assigned at the C-30 position according to the HMBC cross-peak at δ_H_ 0.91 (3H, H-29)/δ_C_ 67.3 (C-30), and the ROESY correlation at δ_H_ 2.33 (1H, dd, *J* = 14.1, 4.1 Hz, H-18)/δ_H_ 3.49 (1H, H-30_a_), 3.51 (1H, H-30_b_).

The configurations of C-3 and C-16 were determined by the correlations observed in the ROESY spectrum between the H-3 α-axial and the H-5 α-axial, and between the H-16 α-axial and the H_3_-27 α-axial, respectively (Figure 2). The α-equatorial orientation of H-22 was established by its low coupling constant value (dd, *J* = 3.5, 1.2 Hz) and by its ROESY connectivity with H-16 α-axial. The aglycone structure of saponin **1** was therefore recognized as 22-*O*-acetylolean-12-ene-3β,16β,22β,30-tetrol. (Appendix A. 1H spectrum of compound **1**; Appendix A. 13C spectrum of compound **1**; Appendix A. HSQC spectrum of compound **1**; Appendix A. HMBC spectra of compound **1**; Appendix A. COSY spectrum of compound **1**; Appendix A. ROESY spectrum of compound **1**.).

In the osidic part of compound **1,** the HSQC spectrum showed three anomeric signals at δ_H_ 4.42 (1H, d, *J* = 7.6 Hz)/δ_C_ 104.3, δ_H_ 4.88 (1H, d, *J* = 7.6 Hz)/δ_C_ 100.7 and δ_H_ 5.19 (1H, br s)/δ_C_ 100.6. The ring protons of the monosaccharide residues were assigned mainly by COSY, TOCSY, HSQC, and HMBC experiments, which allowed the identification of one glucuronopyranosyl (GlcA), one glucopyranosyl (Glc), and one rhamnopyranosyl (Rha) units (Table 1). The large ^3^*J*_H-1,H-2_ values in the ^1^H NMR spectrum of glucuronic acid and glucose in their pyranose form (7.6 Hz) indicated their β anomeric orientation. The large ^1^*J*_H-1,C-1_ value of the Rha (167 Hz) confirmed that the anomeric proton was equatorial (α-pyranoid anomeric form). The absolute configurations of the sugars were determined to be D for GlcA and Glc, and L for Rha (Experimental section). The same protocol was used for the identification of the monosaccharides of compound **2**.

The HMBC correlations at δ_H_ 3.70 (1H, dd, *J* = 9.4, 7.6 Hz, GlcA-2)/δ_C_ 100.7 (Glc-1) and at δ_H_ 3.37 (1H, dd, *J* = 9.4, 7.0 Hz, Glc-2)/δ_C_ 100.6 (Rha 1) suggest an α-L-rhamnopyranosyl-(1→2)-β-D-glucopyranosyl-(1→2)- β-D-glucuronopyranosyl sequence (Table 1, Figure 1). The HMBC correlation at δ_H_ 4.42 (1H, GlcA-1)/δ_C_ 90.4 (C-3) and the ROESY cross-peak at δ_H_ 4.42 (1H, GlcA-1)/δ_H_ 3.22 (1H, H-3_ax_), proved a 3-*O*-heterosidic bond between GlcA and C-3 of the aglycone. Based on the above results, the structure of compound **1** was elucidated as 3-*O*-α-L-rhamnopyranosyl-(1→2)-β-D-glucopyranosyl-(1→2)-β-D-glucuronopyranosyl-22-*O*-acetylolean-12-ene-3β,16β,22β,30-tetrol. (Appendix A. 1H spectrum of compound **2**; Appendix A. 13C spectrum of compound **2**; Appendix A. HSQC spectrum of compound **2**; Appendix A. HMBC spectrum of compound **2**; Appendix A. COSY spectrum of compound **2**; Appendix A. ROESY spectrum of com-pound 2.).

Analysis of the mass spectrum of compound **2** in HRESIMS (positive mode) reveals a quasi-molecular ion at *m*/*z* 803.4195 [M + Na]^+^, in agreement with the molecular formula of C_41_H_64_O_14_Na. This suggests a molecular mass of 780 g/mol.

The ^1^H NMR spectrum part of the aglycon displayed signals attributable to six angular methyl groups (3H, s, each) at δ_H_ 0.88 (H-25), 0.93 (H-29), 0.97 (H-28), 0.97 (H-26), 1.18 (H-23), and 1.24 (H-27), one olefinic proton at δ_H_ 5.38 (1H, br t, *J* = 3.0 Hz, H-12), one secondary alcoholic function at δ_H_ 3.35 (1H, dd, *J* = 11.7, 2.9 Hz, H-3), and two primary ones at δ_H_ 3.23 (1H, d, *J* = 12.0 Hz, H-24_a_), 4.06 (1H, d, *J* = 12.0 Hz, H-24_b_) and δ_H_ 3.28 (1H, d, *J* = 12.9 Hz, H-30_a_), 3.34 (1H, d, *J* = 12.9 Hz, H-30_b_) (Table 1). Additionally, HMBC correlations at δ_H_ 2.18 (1H, d, *J* = 14.4 Hz, H-21_a_)/δ_C_ 218.0 (C-22) and δ_H_ 2.36 (1H, d, *J* = 14.4 Hz, H-21_b_)/δ_C_ 218.0 (C-22), revealed the presence of a ketone function at the 22-position. A deshielded signal corresponding to a 3-*O*-glycosidic linkage can be observed at position 3 of the aglycon at δ_H_ 3.35 (1H, H-3)/δ_C_ 90.3 (C-3). Moreover, the first CH_2_OH group was assigned at the C-30 position as in compound **1**, according to the HMBC cross-peak at δ_H_ 0.93 (3H, H-29)/δ_C_ 67.4 (C-30), and the ROESY correlations at δ_H_ 2.46 (1H, dd, *J* = 13.5, 3.5 Hz, H-18)/δ_H_ 3.28 (1H, H-30_a_), 3.34 (1H, H-30_b_). The second CH_2_OH group was located at the C-24 position by the correlation in the HMBC spectrum at δ_H_ 1.18 (3H, H-23)/δ_C_ 62.5 (C-24), and the ROESY correlations at δ_H_ 4.06 (1H, H-24_b_)/δ_H_ 0.88 (3H, H-25) and δ_H_ 0.97 (3H, H-26). This genin is known as wistariasapogenol A, already described as the aglycon of saponins of *W. brachybotrys* [13].

Analysis of the HSQC spectrum showed the presence of two anomeric protons at δ_H_ 4.45 (d, *J* = 7.6 Hz) and 4.63 (d, *J* = 7.6 Hz), giving correlations with two anomeric carbons at δ_C_ 103.5 and 103.7 ppm, respectively. According to the same protocol as for compound **1**, a β-D-glucuronopyranosyl and a β-D-xylopyranosyl moieties were identified (Table 1, Figure 1). ROESY correlations have been mainly used to establish the structure of the oligosaccharide chain, between δ_H_ 4.45 (1H, GlcA-1)/δ_H_ 3.35 (1H, H-3_ax_) and δ_H_ 4.63 (1H, Xyl-1)/δ_H_ 3.49 (1H, dd, *J* = 9.4, 7.6 Hz, GlcA-2). Therefore, the structure of compound **2** is established as 3-*O*-β-D-xylopyranosyl-(1→2)-β-D-glucuronopyranosylwistariasapogenol A.

Moreover, five additional known compounds were obtained (**3**–**7**) (Figure 1), namely 3-*O*-α-L-rhamnopyranosyl-(1→2)-β-D-xylopyranosyl-(1→2)-β-D-glucuronopyranosywistariasapogenol A (wistariasaponin A) (**3**) [13], 3-*O*-α-L-rhamnopyranosyl-(1→2)-β-D-xylopyranosyl-(1→2)-β-D-glucuronopyranosylsoyasapogenol E (wistariasaponin D) (**4**) [11], 3-*O*-α-L-rhamnopyranosyl-(1→2)-β-D-glucopyranosyl-(1→2)-β-D-glucuronopyranosylsoyasapogenol E (dehydroazukisaponin V) (**5**) [14], 3-*O*-α-L-rhamnopyranosyl-(1→2)-β-D-xylopyranosyl-(1→2)-β-D-glucuronopyranosylsoyasapogenol B (**6**) (astragaloside VIII) [15], 3-*O*-α-L-rhamnopyranosyl-(1→2)-β-D-glucopyranosyl-(1→2)-β-D-glucuronopyranosylsoyasapogenol B (azukisaponin V) (**7**) [14].

The sweet taste properties of the saponins with the higher amounts after purification, compounds **1**–**3**, were evaluated using the stimulation of the human taste heterodimer receptor TAS1R2/TAS1R3, with sucralose as reference (EC_50_ = 16 ± 2 μg/mL). They were applied on a cell-based heterologous expression system and compared to glycyrrhizin. Glycosides **1**–**3** shared a common oleanane-type aglycone with a primary alcoholic function at the 30-position, a 3-*O*-β-D-glucuronopyranosyl linkage. However, only saponins **2** and **3** activated the sweet taste receptor with EC_50_ values at 28 ± 2 μg/mL and 29 ± 7 μg/mL, respectively, both in the same range as glycyrrhizin (EC_50_ = 34 ± 3 μg/mL) (Figure 3). Comparing compound **1** with compounds **2** and **3**, compound **1** possesses a 16β-OH, a 22β-*O*-acetyl, and a free 24β-CH_3_ group, instead of a 22-ketone, a free 16-CH_2_, and a 24β-CH_2_OH function in compounds **2** and **3**.

Structurally, compounds **2** and **3** shared with glycyrrhizin a 3-*O*-β-D-glucuronopyranosyl linkage, a ketone group, and an oxidated 30β-CH_3_ (Figure 1). Previous studies highlighted the key role of the 3-*O*-β-D-glucuronopyranosyl group, the ketone function, and the oxidation of the 24β-CH_3_ and the 30β-CH_3_ groups [12]. These results are very promising since saponins **2** and **3** can activate the sweet-taste receptor TAS1R2/TAS1R3 with EC_50_ values in the micromolar range. These values are close to those measured for sucralose, a sweetener widely used by the food industry, and the glycyrrhizin highly appreciated for its sweetness with a typical licorice taste. However, saponins are known for their toxicity on cell membranes, so, before any further investigations, the toxicity of these molecules has to be evaluated.

## 3. Materials and Methods

### 3.1. General Experimental Procedures

NMR spectra were recorded on a Varian INOVA 600 MHz spectrometer (Agilent Technologies) equipped with 3 mm triple resonance inverse and 3 mm dual broadband probe heads. Spectra were recorded in methanol-*d*_4_, and all spectra were recorded at T = 308.15 K. Pulse sequences were taken from the Varian pulse sequence library (gCOSY; gHSQCAD and gHMBCAD with adiabatic pulses CRISIS-HSQC and CRISIS-HMBC). TOCSY spectra were acquired using DIPSI spin-lock and 150 ms mixing time. Mixing time in ROESY experiments was 300 ms. Chemical shifts were reported in δ units and coupling constants (*J*) in Hz. HR-ESIMS (positive-ion mode) and ESIMS (positive- and negative-ion mode) were carried out on a Bruker micrOTOF mass spectrometer. A MARS 6 microwave apparatus (CEM) was used for the extractions. Isolations of the compounds were carried out using column chromatography (CC) with Sephadex LH-20 (550 mm × 20 mm, GE Healthcare Bio-Sciences AB), and vacuum liquid chromatography (VLC) with reversed-phase RP-18 silica gel (75–200 μm, Silicycle). Medium-pressure liquid chromatography (MPLC) was performed using silica gel 60 (Merck, 15–40 μm) with a Gilson M 305 pump (25 SC head pump, M 805 manometric module), a Büchi glass column (460 mm × 25 mm and 460 mm × 15 mm), and a Büchi precolumn (110 mm × 15 mm). Thin-layer chromatography (TLC, Silicycle) and high-performance thin-layer chromatography (HPTLC, Merck) were carried out on precoated silica gel plates 60 F_254_, solvent system CHCl_3_/MeOH/H_2_O/AcOH (60:32:7:1 and 70:30:5:1). The spray reagent for saponins was vanillin reagent (1% vanillin in EtOH/H_2_SO_4_, 50:1). The HPLC was performed on an Agilent 1260 instrument, equipped with a degasser, a quaternary pump, a sample changer, and a UV detector (210 nm). The chromatographic separation for the analytical part was carried out on a C18 column (250 mm × 4.6 mm internal diameter, 5 μm; Phenomenex LUNA) at room temperature and protected by a guard column. The mobile phase consists of (A) 0.01% (*v*/*v*) aqueous trifluoroacetic acid and (B) acetonitrile delivered at 1 mL/min going from 30% to 80% B in 30 min. The injection volume was 10 µL.

### 3.2. Plant Material

*Wisteria sinensis* was purchased from Botanic^®^ (Quetigny, France) in September 2019, and a sample was deposited in the herbarium at the Laboratory of Pharmacognosy, Université de Bourgogne Franche-Comté, Dijon, France, under the number N° 2019/09/06.

### 3.3. Extraction and Isolation

Microwave-assisted extraction of 47.07 g of dried pulverized roots was carried out three times, with a mixture of EtOH/H_2_O (75:35; 500 mL). The microwave apparatus was programmed to reach 60 °C in 10 min, and then maintain this temperature for another 30 min with moderate agitation. After evaporation of the solvent under vacuum, the resulting extract (6.65 g) was submitted to VLC (RP-18 silica gel, H_2_O, MeOH/H_2_O 50:50, and MeOH). The fraction eluted with 50:50 MeOH/H_2_O (1.44 g) was fractionated by CC (Sephadex LH-20, MeOH), to give a fraction rich in saponins (846 mg), which was further fractionated by MPLC on silica gel 60 (15–40 μm, CHCl_3_/MeOH/H_2_O 70:30:5, 60:32:7, 64:40:8, 2.5 mL/min). Fractions F1-F16 were obtained, and F4 (31.4 mg), F8 (93.0 mg), F9 (103.7 mg) and F11 (137.1 mg) were fractionated again by successive MPLC on silica gel RP-18 (MeOH/H_2_O 25:75 to 40:60 and 100:0), to give compounds **1** (5.2 mg), **2** (4.7 mg), wistariasaponin A (**3**) (20.4 mg), wistariasaponin D (**4**) (3.2 mg), dehydroazukisaponin V (**5**) (3.2 mg), astragaloside VIII (**6**) (4.5 mg), and azukisaponin V (**7**) (4.5 mg).

*3-O-α-L-rhamnopyranosyl-(1→2)-β-D-glucopyranosyl-(1→2)-β-D-glucuronopyranosyl-22-O-acetylolean-12-ene-3β,16β,22β,30-tetrol (***1***)*. White, amorphous powder; ^1^H and ^13^C NMR data (600 MHz and 150 MHz, MeOD), see Table 1; HR-ESIMS (positive-ion mode) *m*/*z* 1023.5051 [M + Na]^+^ (calcd. for C_50_H_80_O_20_Na, 1023.5141).

*3-O-β-D-xylopyranosyl-(1→2)-β-D-glucuronopyranosylwistariasapogenol A (***2***).* White, amorphous powder; ^1^H and ^13^C NMR data (600 MHz and 150 MHz, MeOD), see Table 1; HR-ESIMS (positive-ion mode) *m/z* 803.4195 [M + Na] + (calcd. for C_41_H_64_O_14_Na, 803,4194).

### 3.4. Acid Hydrolysis and Absolute Configuration Determination

An aliquot (150 mg) of a rich saponin fraction was hydrolyzed with 2N aqueous CF_3_COOH (25 mL) for 3 h at 95 °C. After extraction with CH_2_Cl_2_ (3 × 15 mL), the aqueous layer was evaporated to dryness with H_2_O until neutral to give the sugar residue (55 mg). Glucuronic acid, glucose, xylose and rhamnose were identified by comparison with authentic samples by TLC using CH_3_COOEt/CH_3_COOH/CH_3_OH/H_2_O (65:25:15:15). After purification of these sugars by prep-TLC in the same solvent, the optical rotation of each purified sugar was measured as follows: D-glucuronic acid: *R*_f_ = 0.24, [α]^25^_D_ + 15 (*c* 0.2, H_2_O), D-glucose: *R*_f_ = 0.50, [α]^25^_D_ + 110 (*c* 0.2, H_2_O), D-xylose: *R*_f_ = 0.60, [α]^25^_D_ + 85 (*c* 0.2, H_2_O), and L-rhamnose: *R*_f_ = 0.69, [α]^25^_D_ + 10 (*c* 0.2, H_2_O).

### 3.5. Bioactivity Assay

We investigated the ability of purified compounds to activate the sweet taste receptor using heterologous expression of TAS1R2/TAS1R3 and functional calcium imaging. As previously described, the cDNAs for TAS1Rs and the plasmid pGP-CMV-GCaMP6S (Addgene #40753) coding for a calcium biosensor, were transiently transfected into HEK293T cells stably expressing the chimeric G-protein subunit Gα16gust44, using Fugene HD (Promega) [6]. Cells transfected only with calcium indicator vector served as negative control. Prior to the stimulation, the transfected cells were washed with C1 buffer (130 mM NaCl, 5 mM KCl, 10 mM Hepes, 2 mM CaCl_2_, 5 mM sodium pyruvate, pH 7.4). Then, we monitored calcium mobilization following sweet taste receptor activation after automatic injection of test substances with a Molecular Devices FlexStation 3 system. Compounds **2** and **3** were dissolved first at 10 mg/mL in DMSO with good solubility. Further dilutions were prepared in C1 solution. The compounds **2** and **3** were evaluated up to a maximum range of 50 µg/mL, because they elicited non-specific calcium responses in mock-transfected cells at concentration above 100µg/mL.
Data were collected from at least three independent experiments carried out in duplicate. The concentration-response curves were obtained after correction of calcium signals for the response of mock transfected cells and normalization to the fluorescence of cells prior to the stimulation. EC_50_ values were calculated using a four-parameter logistic nonlinear regression with equation [f(x) = min + (max–min)/(1 + (x/EC_50_)^nH^)] with curves fitting of Sigma Plot software.

## Figures and Tables

**Figure 1 molecules-27-07866-f001:**
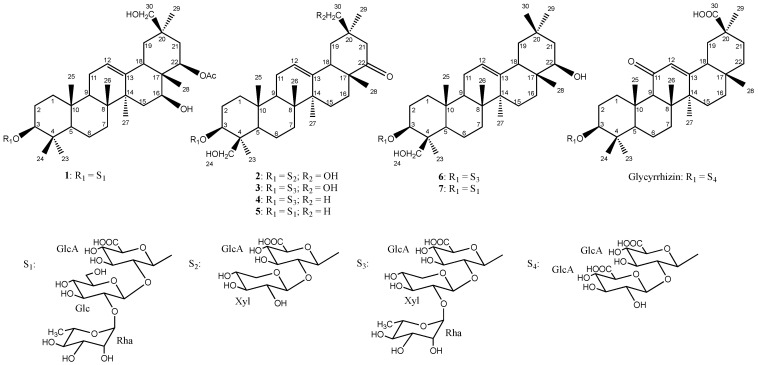
Structures of compounds **1**–**7** and glycyrrhizin.

**Figure 2 molecules-27-07866-f002:**
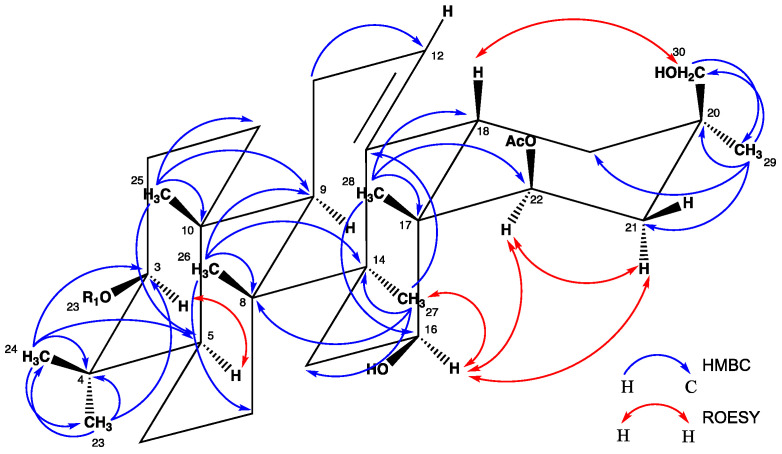
Key HMBC and ROESY correlations for the aglycone of **1**.

**Figure 3 molecules-27-07866-f003:**
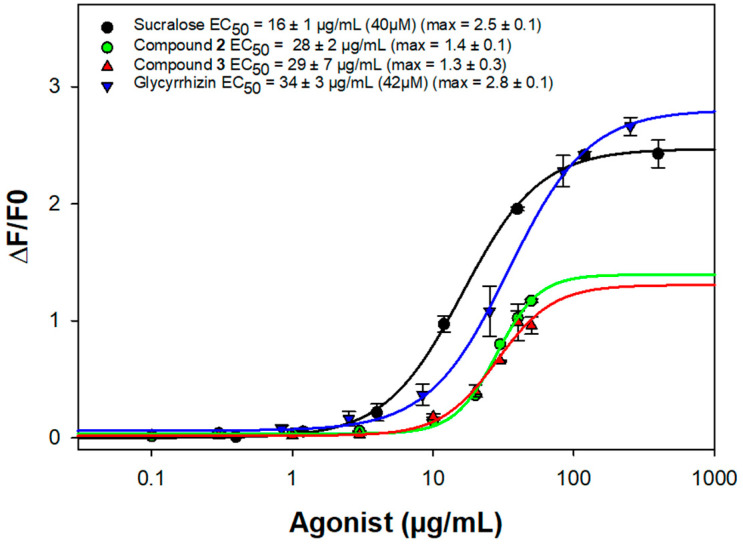
Activation of TAS1R2/TAS1R3 by sucralose, compound **2**,**3** and glycyrrhizin.

**Table 1 molecules-27-07866-t001:** ^13^C NMR and ^1^H NMR data of compounds **1**-**2** in MeOD (δ ppm, *J* in Hz) ^1^.

Position	1		2	
	δ_C_	δ_H_ (*J* in Hz)	δ_C_	δ_H_ (*J* in Hz)
1	38.5	1.02 m 1.64 m	38.4	1.02 m 1.63 m
2	25.4	1.72 m 2.13 m	25.5	2.10 m 2.13 m
3	90.4	3.22	90.3	3.35 dd (11.7, 2.9)
4	39.1	-	43.3	-
5	55.5	0.80	56.0	0.95
6	17.9	1.44 m 1.59 m	18.0	1.41 m 1.66 m
7	32.4	1.37 m 1.59 m	32.6	1.40 m 1.58 m
8	39.8	-	39.4	-
9	46.8	1.57 m	47.8	1.62 m
10	36.3	-	36.1	-
11	23.2	1.91 m 1.93 m	23.4	1.90 m 1.92 m
12	123.0	5.33 br t (3.0)	123.8	5.38 br t (3.0)
13	142.3	-	141.4	-
14	42.9	-	41.6	-
15	35.9	1.31 m 1.76 m	24.8	1.11 m 1.79 m
16	66.2	4.15 dd (11.7, 4.7)	26.7	1.21 m 2.07 m
17	41.0	-	46.9	-
18	45.7	2.33 dd (14.1, 4.1)	46.9	2.46 dd (13.5, 3.5)
19	40.5	1.34 1.73	41.8	1.62 1.99
20	34.5	-	38.6	-
21	32.8	1.49 1.75	46.0	2.18 d (14.4) 2.36 d (14.4)
22	74.2	5.24 dd (3.5, 1.2)	218.0	-
23	15.4	0.88 s	21.3	1.18 s
24	27.3	1.11 s	62.5	3.23 d (12.0) 4.06 d (12.0)
25	14.7	0.98 s	14.7	0.88 s
26	16.1	1.02 s	16.0	0.97 s
27	26.5	1.26 s	24.3	1.24 s
28	13.8	0.79 s	19.9	0.97 s
29	27.0	0.91 s	25.4	0.93 s
30	67.3	3.49 d (14.1) 3.51 d (14.1)	67.4	3.28 d (12.9) 3.34 d (12.9)
at C-22				
COCH_3_	171.1	-		
CH_3_CO	20.0	2.02 (3H, s)		
GlcA-1	104.3	4.42 d (7.6)	103.5	4.45 d (7.6)
2	79.6	3.70 dd (9.4, 7.6)	79.6	3.49 dd (9.4, 7.6)
3	77.3	3.60 t (9.4)	76.7	3.61 t (9.4)
4	72.6	3.42 t (9.4)	72.1	3.48
5	75.1	3.52	75.3	3.60
6	175.3	-	175.7	-
Xyl-1			103.7	4.63 d (7.6)
2			74.1	3.19 dd (9.4, 7.6)
3			77.0	3.28 t (9.4)
4			69.4	3.49 m
5			65.5	3.12 dd (11.1, 9.4) 3.81 dd (11.1, 5.3)
Glc-1	100.7	4.88 d (7.6)		
2	78.2	3.37 dd (9.4, 7.6)		
3	77.9	3.44 t (9.4)		
4	71.3	3.05 t (9.4)		
5	76.7	3.22 m		
6	62.2	3.82 dd (11.7, 2.4) 3.52 dd (11.7, 7.6)		
Rha-1	100.6	5.19 br s		
2	70.9	3.91 br s		
3	70.8	3.75 dd (9.4, 2.9)		
4	72.8	3.39 t (9.4)		
5	68.2	4.12 dq (9.4, 5.9)		
6	16.9	1.25 d (5.9)		

^1^ Overlapped signal are reported without designated multiplicity. δ in ppm; *J* in parentheses in Hz.

## Data Availability

Not applicable.

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
