# Peer review of "Activation of a Sweet Taste Receptor by Oleanane-Type Glycosides from Wisteria sinensis"

_molecules, 2022, doi:10.3390/molecules27227866_

Round 1

Reviewer 1 Report

The aim of MS presented by Hobloss, S., et al. was focused on the isolation of oleanane-type glycosides from Wisteria sinensis and the assessment of their potential use as artificial sweeteners. Authors identified two compounds that were able to activate TAS1R2/TAS1R3 receptors, with similar way than two non-caloric sugars, the glycyrrhizin and sucralose. The work in interesting and well-presented, however some points should be addressed to improve the MS.

1.      The last phrase of introduction section (lines 43-45) must specify the name of compounds.

2.     Include the molecular mass units of the compounds in lines 56 and line 108.

3.     It is desirable to write “compounds”, “saponin”, “molecule”, or any other subject when before the number of the oleanane-type glycosides identified.

E.G. Line 83: In the osidic part of 1, the HSQC spectrum…

Should be write: In the osidic part of compound 1, the HSQC spectrum…

The same for lines 93, 99, 153, 155, 156, 157, 163.

4.     I suggest moving the glycyrrhizin structures to figure 1, in order to compare with the ones were reported in the work.

5.     Figure 3. Authors must to test higher concentrations of compounds 2 and 3. In order to obtain a reliable EC50, the curve must to include some points at Rmax, which was not reached in this assay. It is clear that the assay shows the activation of TAS1R2/TAS1R3 receptors by compounds 2 and 3, however is not possible to estimate EC50 when Rmax is missing.

Reviewer 2 Report

The article is devoted to isolation oleanane-type glycosides from Wisteria sinensis as well as investigation of their sweet taste properties. Two of the seven isolated glycosides were new. Structures of the compounds are reliably proven by spectral methods. I recommend the article for publication, A few comments are listed below.

1. Abstract, lines 12-13: "After successive purifications by various chromatographic methods, their structures were elucidated by an extensive 600 MHz" Is this information necessary for the abstract? I don't think so.

2. Table 1. columns δH (J en Hz): should it be δH (J in Hz)?

3. line 95: "suggest a α-L-rhamnopyranosyl" should be"suggest an α-L-rhamnopyranosyl"
